# Alginate Oligosaccharides Repair Liver Injury by Improving Anti-Inflammatory Capacity in a Busulfan-Induced Mouse Model

**DOI:** 10.3390/ijms24043097

**Published:** 2023-02-04

**Authors:** Yanan Hao, Hanhan Fang, Xiaowei Yan, Wei Shen, Jing Liu, Pengfei Han, Yong Zhao, Weidong Zhang, Yanni Feng

**Affiliations:** 1Laboratory of Animal Reproductive Physiology and Disease, College of Veterinary Medicine, Qingdao Agricultural University, Qingdao 266109, China; 2College of Life Sciences, Qingdao Agricultural University, Qingdao 266109, China; 3College of Science, Health, Engineering and Education, Murdoch University, Perth 6150, Australia; 4State Key Laboratory of Animal Nutrition, Institute of Animal Sciences, Chinese Academy of Agricultural Sciences, Beijing 100193, China; 5Core Laboratories of Qingdao Agricultural University, Qingdao 266109, China

**Keywords:** liver diseases, AOS, blood metabolites, inflammation

## Abstract

Liver diseases are associated with many factors, including medicines and alcoholics, which have become a global problem. It is crucial to overcome this problem. Liver diseases always come with inflammatory complications, which might be a potential target to deal with this issue. Alginate oligosaccharides (AOS) have been demonstrated to have many beneficial effects, especially anti-inflammation. In this study, 40 mg/kg body weight (BW) of busulfan was intraperitoneally injected once, and then the mice were dosed with ddH_2_O or AOS 10 mg/kg BW every day by oral gavage for five weeks. We investigated AOS as a potential no-side-effect and low-cost therapy for liver diseases. For the first time, we discovered that AOS 10 mg/kg recovered liver injury by decreasing the inflammation-related factors. Moreover, AOS 10 mg/kg could improve the blood metabolites related to immune and anti-tumor effects, and thus, ameliorated impaired liver function. The results indicate that AOS may be a potential therapy to deal with liver damage, especially in inflammatory conditions.

## 1. Introduction

Liver diseases affect approximately 300 million people in China, and around 30% of the US population [1,2]. Many causative factors drive liver diseases, including diet, alcohol abuse, viral infection, and auto-immune diseases [3,4]. Alcoholic liver disease (ALD) and chronic liver diseases are related to molecular patterns that stimulate inflammatory signals, oxidative damage, and necrosis to develop into fibrosis and cirrhosis [5,6]. Moreover, non-alcoholic fatty liver disease (NAFLD) has been observed to be related to inflammation, which plays a vital role in hepatic fibrogenesis, which is correlated with a high prevalence of metabolic ailments [7,8,9,10,11]. AOS, the degradation products of alginate (one type of marine polysaccharide from brown seaweed), are composed of α-L-guluronate (G) and β-D-mannuronate (M) joined by 1, 4-glycoside bonds [12]. They are natural products with attractive pharmaceutical properties that exert many beneficial effects [13,14]. They can downregulate the expression of pro-inflammatory cytokines, decrease macrophage infiltration, and upregulate the expression of anti-inflammatory cytokines, which possess anti-oxidative and anti-inflammatory effects [15,16,17,18]. However, the ability of AOS to protect against liver injury has not been reported yet.

Busulfan is usually used to treat various hematologic malignancies, immunodeficiencies, and chronic myelogenous leukemia in children and adults. It is one of the most widely used chemotherapeutics in the allogeneic hematopoietic stem cell transplantation (HSCT) conditioning regimen [19,20]. However, many adverse side effects have been reported since it is a cytostatic drug that consumes up to 60% of hepatic glutathione (GSH), an important intracellular antioxidant [21]. Busulfan could induce systemic histopathological changes in the heart, lungs, stomach, intestines, liver, kidneys, and testes [19]. Therefore, we used busulfan to establish the liver injury model in the current study.

Our previous study found that AOS improves the intestinal barrier function by benefiting intestinal cell growth, cell–cell junctions, and gut microbiota, and rescuing busulfan-diminished male infertility by improving germ cell development and the testicular microenvironment [22,23]. However, it is unknown whether AOS can improve liver function and its mechanisms. Therefore, the current study aimed to explore the beneficial improvement of AOS on busulfan-disrupted mouse liver and the underlying mechanisms.

## 2. Results

### 2.1. AOS Repaired Busulfan-Induced Liver Injury

Previously, we found that AOS 10 mg/kg BW did not affect the blood metabolism, testis, or other organs. In the current study, the mouse livers showed tight cell connections, a complete hepatic sinus structure, and a dense cytoplasmic matrix, and the samples in the A10 group were similar to those in the A0 group, Next, we focused on three groups: A0, BA0, and BA10. The busulfan caused liver cellular damage, as shown by TEM analysis (Figure 1b). In the BA0 group mouse livers, the hepatocytes were loosely arranged, and hepatic sinuses were dilated. Moreover, the mitochondria showed swelling and deformation, mitochondrial rupture, mitochondrial autophagy, matrix reduction, and matrix vacuolation in the BA0 mouse liver samples. Interestingly, compared to that in the BA0 group, the liver cells were arranged more closely, the expansion of hepatic sinuses was reduced, the cell matrix was rich, and the glycogen granules were increased in the BA10 group. Furthermore, similar to that in the A0 group, rough endoplasmic reticulum was abundant, and the membrane was clear and complete in the BA10 group. The data indicate that AOS 10 mg/kg (BA10) could repair busulfan-induced liver cell injury. At the same time, B0 increased the liver damage markers alanine aminotransferase (ALT) and aspartate aminotransferase (AST) in the mouse blood samples, while BA10 decreased them to the same levels as in the control (A0) group (Figure 1c,d). Furthermore, compared to the A0 group, the protein levels of the liver injury markers signal transducer and activator of transcription 3 (STAT3) and tribbles pseudokinase 3 (TRIB3) were increased by busulfan (BA0 group) while they were decreased in the BA10 group (Figure 2). The data together suggest that AOS had the capacity to repair busulfan-induced liver damage.

Liver injury usually accompanies oxidative stress, and AOS is known to have good antioxidant activity [24]. Busulfan increased the protein levels of superoxide dismutase 1 (SOD1) in the liver, which was decreased by BA10 (Figure 3a,b). Moreover, glutathione peroxidase 1 (GPX1) is a critical factor of anti-oxidation. The protein level of GPX1 was increased by busulfan, while it was decreased by AOS (Figure 3c,d). The data indicate that busulfan induced oxidative stress, and the liver cells increased SOD1 and GPX1 to reduce it. However, AOS reduced the oxidative stress, and SOD1 and GPX1 were not changed compared to the control group (A0 group).

### 2.2. AOS Improved Busulfan-Disturbed Gene Expression in Liver

To explore how AOS improved busulfan-induced liver injury, the liver gene expression was determined by RNA-seq analysis, which showed that busulfan downregulated the expressions of 749 genes and upregulated the expressions of 128 genes, while BA10 decreased the expressions of 121 genes and increased the expressions of 191 genes (Figure 4a). The functional enrichment analysis discovered that cell proliferation- and immune-related pathways were enriched for the genes that were decreased in BA0 and increased in BA10; however, the inflammation-related pathways were enriched for the genes that were increased in BA0 and decreased in BA10 (Figure 4b,c). The q-RT-PCR data confirm the gene expression patterns (Figure 4d). Moreover, one of the important gene early growth responses (Egr1) for cell growth and proliferation was significantly increased by BA10 (Figure 4d). The data suggest that busulfan disturbed liver cell gene expression, causing liver damage, while AOS could ameliorate the gene expression to repair busulfan-induced liver injury. Moreover, the data indicate that inflammation may be involved in the process.

### 2.3. AOS Diminished Busulfan-Induced Inflammation Status in Liver

The RNA-seq data indicate that inflammation might be involved in busulfan-induced liver injury. The protein levels of interferon γ (IFN-γ), interleukin-6 (IL-6), interleukin-6 receptor (IL-6R), interleukin-22 receptor (IL-22R), transforming growth factor-β 1 (TGF-β1), and tumor necrosis factor-α (TNF-α) were increased in the BA0 group compared to the A0 group by immunofluorescence (IHF) staining and WB (Figure 5). Compared to A0, the protein levels of these inflammatory factors were increased in BA0 but decreased in BA10 (Figure 5). However, the protein levels of the inflammation-related factors were reduced in the BA10 group compared to the BA0 group (Figure 5). The data suggest that AOS 10 mg/kg may repair busulfan-induced liver damage by suppressing the inflammation-related signal pathways.

### 2.4. AOS Benefitted the Blood Metabolism to Support the Repair of Busulfan-Induced Liver Damage

In the current study, we found that AOS ameliorated busulfan-induced liver damage by improving the blood metabolites (Appendix A). Compared to A0, 814 metabolites were increased, while 854 metabolites were decreased by BA0. Compared to BA0, 838 metabolites were elevated, while 830 were reduced by BA10 (Figure 6a). In the changed metabolites, there was a clear trend for organic acid and derivatives, which were decreased in the BA0 group and increased in the BA10 group (Figure 6b). Some metabolites have been found to be associated with liver inflammatory reactions and antioxidant capacities, such as tuftsin, melanostatin, S-cysteinosuccinic acid, and L-tyrosine [25,26,27,28]. These metabolites were decreased in the BA0 group but increased in the BA10 group (Figure 6c). Moreover, these metabolites were negatively correlated with liver injury-related proteins and inflammation-related factors (Figure 6d). The data further suggest that AOS could improve the systemic environment to ameliorate busulfan-caused liver damage.

## 3. Discussion

Liver diseases are a global health issue and influence millions of people worldwide [1,2]. The liver is the primary organ of complex immunological activity mediated by a diverse immune cell repertoire and non-hematopoietic cell population, and it also is an important metabolism organ in mammals [29]. Constantly changing metabolic and tissue remodeling activity could result in persistent, regulated inflammation [29,30,31]. AOS were demonstrated to have many advanced benefits [23]. However, little is known about the effects of AOS on the liver, especially on the improvement of liver inflammatory injury. Busulfan could increase the risk of liver injury in multivariable analyses and induce liver inflammation [32]. In the current study, we found that AOS 10 mg/kg could ameliorate the liver structure disrupted by busulfan, based on TEM ultrastructural pathological analysis. We discovered that busulfan-induced loosely arranged hepatocytes, dilated hepatic sinuses, and abnormal mitochondrial physiological structure, while AOS could improve busulfan-induced liver damage significantly. Furthermore, RNAseq analysis determined that AOS 10 mg/kg could improve the expressions of genes related to liver function. These data confirm that AOS protects hepatic health.

Busulfan induces liver damage via inflammation, oxidative stress, and metabolic and liver injury-related signaling pathways. In the current study, busulfan increased the protein levels of the genes in pathways related to inflammation (IFN-γ, IL-6, IL-6R, IL-22R, TGF-β, TNF-α) and liver injury (STAT3, TRIB3; SOD1, GPX1), while AOS decreased the protein levels of them significantly. IFN-γ plays a crucial role in activating cellular immunity and stimulating an anti-tumor immune response [33,34]. IL-6 is produced in response to infections and tissue injuries in the liver generation, contributing to host defense by stimulating acute-phase responses, hematopoiesis, and immune reactions [35]. IL-6R is the receptor of IL-6, which regulates inflammatory reactions [36]. Furthermore, the IL-22/IL-22R signaling pathway plays a vital role in hepatocytes in inducing the production of chemokines, and the chemotactic immune cells reach the local area to aggravate the inflammatory pathology [37]. It is known that TGF-β1 is related to cell immunity [38,39]. Moreover, TNF-α is one of the pro-inflammatory cytokines that can cause inflammation and apoptotic cell death and can mediate the release of a variety of cytokines, such as IL-6, interleukin-8 (IL-8), and interleukin-1β (IL-1β), by stimulating macrophages [40,41,42]. It is known that STAT3 plays a pivotal role in the pathogenesis of liver diseases and contributes to the development and progression of fibrosis in the liver, a transcriptional factor involved in immune responses, inflammation, and tumorigenesis [43,44]. Furthermore, TRIB3 is a stress protein upregulated in response to multiple stressors; its elevated expression positively correlates with the development of human hepatic fibrosis associated with suppressed autophagy [45,46]. Oxidative stress always plays a vital role in liver diseases. SOD is an antioxidant metalloenzyme in organisms and plays a vital role in balancing oxidation and anti-oxidation [47]. Our data suggest that AOS could reduce inflammation and liver injury induced by busulfan.

Liver inflammation is closely related to metabolism [48]. Some metabolites interact with liver injury. Tuftsin is a bioactive peptide fragment that promotes phagocytosis associated with the immune system. It can enhance the phagocytosis and pinocytosis of these cells [25]. Our study found that tuftsin was decreased by busulfan but increased by AOS 10 mg/kg. It is a tetra peptide substance related to anti-inflammatory, antioxidant, and anti-tumor pathways. Interestingly, busulfan also decreased some metabolites that were increased by AOS 10 mg/kg, such as melanostatin, S-cysteinosuccinic acid, and L-tyrosine, which regulate the immune system [26,27,28]. AOS could potentially repair the busulfan-induced liver damage by modulating plasma metabolites related to anti-inflammation.

## 4. Materials and Methods

### 4.1. Study Design and Ethics

The animal procedures used in this study were approved by the Committee on the Ethics of Animal Experiments of Qingdao Agricultural University’s Institutional Animal Care and Use Committee (IACUC) (QAUR2020106) [49]. The study was performed according to the recommendations in the Guide for the Care and Use of Laboratory Animals of the National Institutes of Health. Mice were housed in an 8 pm to 8 am light/dark cycle and in a temperature-controlled environment of 23 °C with humidity of 50–70%. In the study, the AOS were donated from Qingdao Bozhihuili CO., LTD., Qingdao, China. Experiments were carried out using three-week-old ICR male mice. The mice were randomly divided into four groups with 30 mice in each group: (1) a control group dosed with ddH_2_O (A0); (2) an AOS 10 mg/kg BW group (A10); (3) a busulfan group dosed with ddH_2_O (BA0), and (4) a busulfan + AOS 10 mg/kg BW group (BA10) (Figure 1a). The busulfan was intraperitoneally injected with 40 mg/kg BW once. The next day, the mice were dosed with ddH_2_O or AOS 10 mg/kg BW every day by oral gavage. The gavage volume was 0.1 mL/mouse/day, and it was administered every morning for five weeks. The AOS was dissolved in ddH_2_O as a stock solution, and then the working dosing solution was freshly diluted in ddH_2_O every day for five weeks. Subsequently, the mice were humanely euthanized to collect the samples for different analyses.

### 4.2. Ultrastructural Pathological Determination by Transmission Electron Microscope (TEM)

The sample preparation for TEM analyses was performed as previously described [50]. Briefly, the mouse livers were collected and fixed in 2% glutaraldehyde made in sodium phosphate buffer (PH 7.2) for 2 h. The specimens were washed extensively to remove excess fixative and subsequently post-fixed in 1% OsO_4_ for one hour in the dark. After extensive washing in phosphate buffer, the samples were dehydrated in an increasingly graded series of ethanol and infiltrated with an increasing concentration of Spur’s embedding medium in propylene epoxide. Then, the specimens were polymerized in an embedding medium at 37 °C for 12 h, at 45 °C for 12 h, and at 60 °C for 48 h. Fifty manometer sections were cut on a Leica Ultracut E equipped with a diamond knife (Diatome, Hatfield, PA, USA) and collected on Formvar-coated, carbon-stabilized molybdenum (Mo) grids. The sections containing grids were stained with uranyl acetate, left to air-dry overnight, and imaged on a JEM-2010F TEM (JEOL Ltd., Tokyo, Japan). Damage in the samples was confirmed using an X-Max^N^80 TLE EDS (Oxford Instruments, Abingdon, UK).

### 4.3. Gene Expression Determined by Quantitative Transcriptomics (RNA-Seq 168 Transcript Profiling) and Messenger RNA (mRNA) Quantitative Real-Time Polymerase Chain Reaction (q-RT-PCR)

#### 4.3.1. RNA Sequencing (RNA-Seq) Analysis

Transcriptomics were analyzed as described in our recent paper [50]. In brief, the total ribonucleic acid (RNA) in livers was isolated using TRIzol Reagent (Invitrogen, Waltham, MA, USA) and purified using RT2 qPCR-Grade RNA Isolation Kit from SABiosciences Co., Ltd. (Frederick, MD, USA). The quality of the RNA was controlled by the A260/A280 ratio being N2.0 and confirmed by electrophoreses, with a fraction of each total RNA sample with sharp 18S and 28S ribosomal RNA (rRNA) bands, as reported in our recent publication [51]. The total RNA samples were first treated with DNase I to degrade any possible deoxyribonucleic acid (DNA) contamination. The mRNA was then enriched using oligo (dT) magnetic beads. The mRNA was reduced to short fragments (about 200 bp). Subsequently, the first strand of cDNA was synthesized using a random hexamer primer. Buffer, dNTPs, RNase H, and DNA polymerase I were added to synthesize the second strand. The double-strand cDNA was purified with magnetic beads, and the 3’-end single nucleotide A (adenine) addition was then performed. Finally, sequencing adaptors were ligated to the fragments. The fragments were enriched by PCR amplification. Agilent 2100 Bioanalyzer and ABI StepOnePlus Real-Time PCR System were used to qualify and quantify the sample library during the QC step. The library products were ready for sequencing via Illumina HiSeqTM 2000. The reads were mapped to reference genes using SOAPaligner (Version 2.20) with a maximum of a two-nucleotide mismatch allowed at the parameters of “-m 0 -x 1000 -s 40 -l 35 -v 3 -r 2”. The number of reads for each gene was transformed into reads per kilo bases per million reads (RPKM), and then differently expressed genes were identified by the DEGseq package using the MA-plot-based method with the random 186 sampling model (MARS) method. The threshold was set as a false discovery rate (FDR) of ≤0.001, and an absolute value of log2Ratio ≥ 1 was set to judge the significance of the difference in gene expression. The data were then analyzed by gene ontology (GO) enrichment and the Kyoto Encyclopedia of Genes and Genomes (KEGG) enrichment.

#### 4.3.2. mRNA q-RT-PCR Analysis

The total RNA was isolated as described above. The total RNA was quantified using a Nanodrop 3300 (ThermoScientific, Waltham, MA, USA) [51]. Two micrograms of the total RNA were used to make the first strand of complementary DNA (cDNA; in 20 µL) using an RT2 First Strand Kit (Cat. No: AT311-192 03, Transgenic Biotech, Beijing, China) following the manufacturer’s instructions. The generated first-strand cDNAs (20 µL) were diluted to 120 µL with double-deionized water (ddH_2_O). Then, 1 µL was used for one PCR reaction (in a 96-well plate). Each PCR reaction (12 µL) contained 6 µL of qPCR Master Mix (Roche, Mannheim, Germany), 1 µL of diluted first-strand cDNA, 0.6 µL primers (10 µM), and 4.4 µL of ddH_2_O. The primers for the qPCR analysis were synthesized by Invitrogen and are presented in Table 1. The qPCR was conducted with the Roche LightCycler^@^ 480 (Roche, Germany) with the following program: Step 1: 95 °C, 10 min; Step 2: 40 cycles of 95 °C, 15 s; 60 °C, 1 min; Step 3: dissociation curve; Step 4: cool down. Three or more independent experimental samples were analyzed.

### 4.4. Histopathological Analysis

The liver was soaked in 4% paraformaldehyde for no less than 4 h. After gradient dehydration, the liver was embedded in paraffin. Liver sections (5 μm) were prepared and subjected to immunostaining as previously described [52]. In brief, the sections were rehydrated with gradient and first blocked with normal goat serum in tris-buffered saline (TBS), followed by incubation (1:150 in TBS-1% BSA) with primary antibodies at 4 °C overnight. After a quick wash with tris–salt buffer with Tween-20 (TBST) (TBS-1‰ Tween 20, 10 min ×3), the sections were incubated with a Cy3-labeled goat anti-rabbit or goat anti-mouse secondary antibody (Beyotime Institute of Biotechnology, Shanghai, China) at 37 °C for 30 min. After three washes with TBST and counterstaining with Hoechst 33342, the stained sections were visualized using a NikonEclipseTE2000-U fluorescence microscope (Nikon, Inc., Melville, NY, USA). Finally, the captured fluorescence images were analyzed using Image J software.

### 4.5. Western Blotting (WB)

The procedure for WB analysis is reported in our previous publications [49]. Briefly, the liver tissue samples were lysed in radioimmunoprecipitation assay (RIPA) buffer containing a protease inhibitor (PI) cocktail from Sangon Biotech, Ltd. (Shanghai, China). The protein concentration was determined using a BCA kit (Beyotime Institute of Biotechnology, Suzhou, China). The primary antibodies used in this study are summarized in Table 2. Actin was used as the loading control. Secondary donkey anti-goat Ab (Cat no.: A0181) was purchased from the Beyotime Institute of Biotechnology, and goat anti-rabbit (Cat no.: A24531) Abs were bought from Novex^®^ by Life Technologies (Carlsbad, CA, USA). Fifty micrograms of total protein per sample were loaded onto 10% sodium dodecyl sulfate (SDS) polyacrylamide electrophoresis gels. The gels were transferred to a polyvinylidene fluoride (PVDF) membrane at 300 mA for two hours at 4 °C. Then, the membranes were blocked with 5% BSA for one hour at RT, followed by three washes with 0.1‰ Tween-20 in TBS (TBST). The membranes were then incubated with primary Abs diluted with 1:500 in TBST with 1% BSA overnight at 4 °C. After three washes with TBST, the blots were incubated with the HRP-labelled secondary goat anti-rabbit or donkey anti-goat Ab, respectively, for one hour at room temperature. After three washes, the blots were imaged. The bands were quantified using Image J software. The intensity of the specific protein band was normalized to actin first, and then the data were normalized to the control.

### 4.6. Plasma Metabolite Measurements by LC/MS

Plasma samples were collected and immediately stored at −80 °C. Before liquid chromatograph mass spectrometer (LC-MS)/MS analysis, the samples were thawed on ice and processed to remove proteins. Then, the samples were detected by ACQUITY UPLC and AB Sciex Triple TOF 5600 (LC/MS) [22]. Fifteen samples in each group were analyzed for the plasma samples.

### 4.7. Statistical Analysis

The data were analyzed using SPSS statistical software (IBM Co., New York, NY, USA) with one-way analysis of variance (ANOVA) followed by least-significant difference (LSD) multiple comparison tests. All groups were compared with each other for every parameter. The data are shown as the mean ± SEM. Statistical significance is based on *p* < 0.05.

## 5. Conclusions

This investigation demonstrates that AOS 10 mg/kg could improve busulfan-injured liver function. The study presents a new idea for treating liver disease, especially with inflammation complications. As a natural product with many physical benefits and no side effects, AOS is a potential therapeutic method to boost liver function for patients with liver diseases. We have reason to believe that AOS and even its derivatives will be widely used in our life and play important roles widely. The underlying mechanisms of AOS are attracting more and more attention.

## Figures and Tables

**Figure 1 ijms-24-03097-f001:**
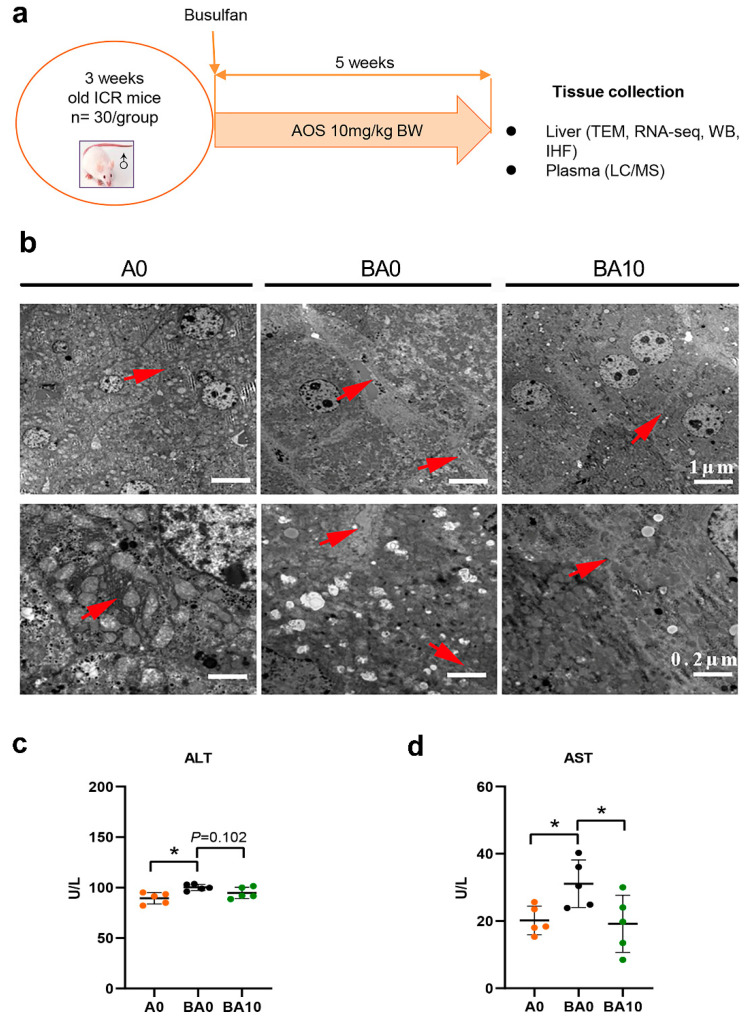
AOS 10 mg/kg improved busulfan-injured liver structure. (**a**). Experimental design (n = 30/group). (**b**). TEM photos with different magnifications. The red arrow indicates the damaged structure of liver. Scale bar: 1 μm/0.2 μm. (**c**). ALT in mouse blood. The *y*-axis represents the concentration of ALT in the mouse blood. The *x*-axis represents the treatment. * *p <* 0.05. (**d**). AST in mouse blood. The *y*-axis represents the concentration of AST in the mouse blood. The *x*-axis represents the treatment. * *p <* 0.05.

**Figure 2 ijms-24-03097-f002:**
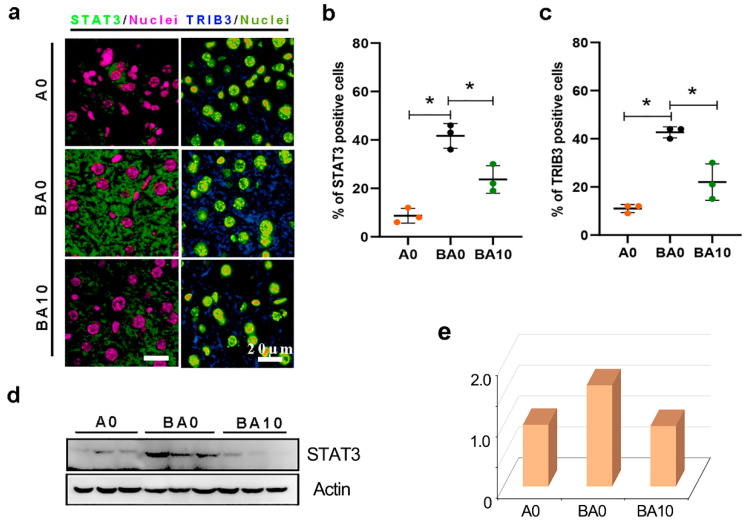
AOS 10 mg/kg improved liver impairment. (**a**). IHF staining of liver injury marker STAT3, TRIB3 in each treatment. Scale bar: 20 μm. (**b**,**c**). Statistical analysis of IHF results. The *y*-axis represents the positive rate of antibodies. The *x*-axis represents the treatment. * *p <* 0.05. (**d**,**e**). Western blotting analysis of liver injury marker STAT3 in each treatment.

**Figure 3 ijms-24-03097-f003:**
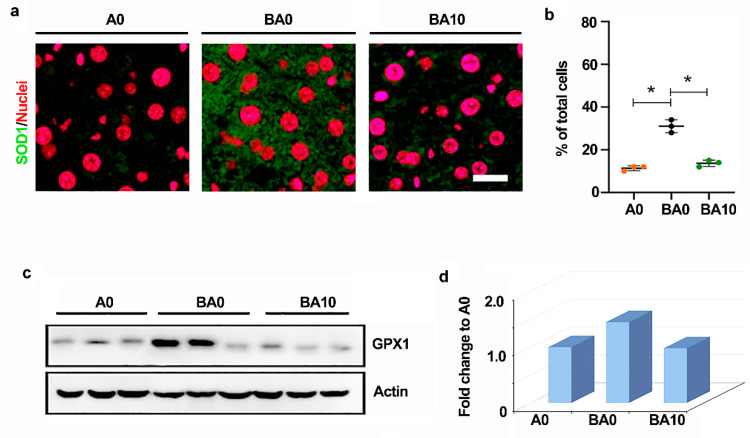
AOS 10 mg/kg benefited antioxidant capability and cell proliferation. (**a**). IHF staining of antioxidant marker SOD1. Scale bar: 20 μm. (**b**). Statistical analysis of IHF results. The *y*-axis represents the positive rate of antibodies. The *x*-axis represents the treatment. * *p <* 0.05. (**c**,**d**). Western blotting analysis of liver antioxidant capability (GPX1) in each treatment.

**Figure 4 ijms-24-03097-f004:**
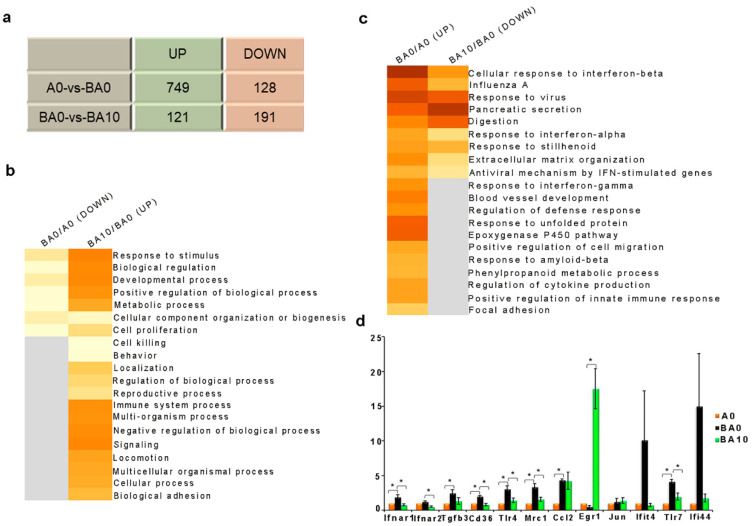
AOS 10 mg/kg repaired busulfan-disrupted liver genes. (**a**). Summary of genes changed in the different compared groups. (**b**,**c**). Gene functional enrichment analysis of genes that were decreased by busulfan and increased by AOS 10 mg/kg based on the RNA-seq analysis of liver tissue by online Metascape software (https://metascape.org/gp/index.html#/main/step1 accessed on 6 July 2020). (**d**). Results of q-RT-PCR. * *p <* 0.05.

**Figure 5 ijms-24-03097-f005:**
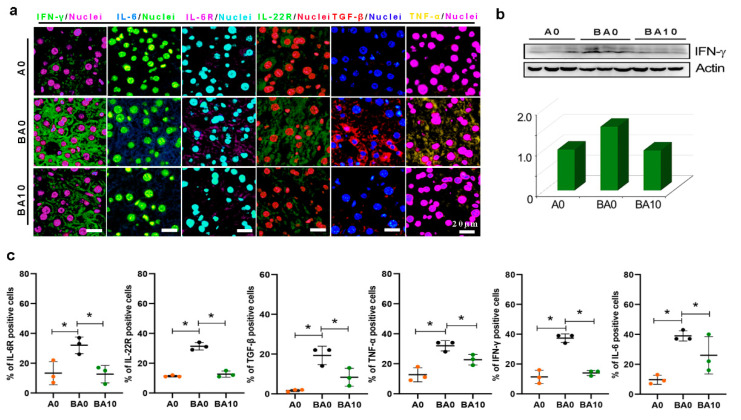
AOS 10 mg/kg ameliorated busulfan-induced liver inflammatory reaction. (**a**). IHF of inflammation-related proteins. Scale bar: 20 μm. (**b**). Western blotting analysis of IFN-γ in each treatment. (**c**). Statistical analysis of IHF. The *y*-axis represents the positive rate of antibodies. The *x*-axis represents the treatment. * *p <* 0.05.

**Figure 6 ijms-24-03097-f006:**
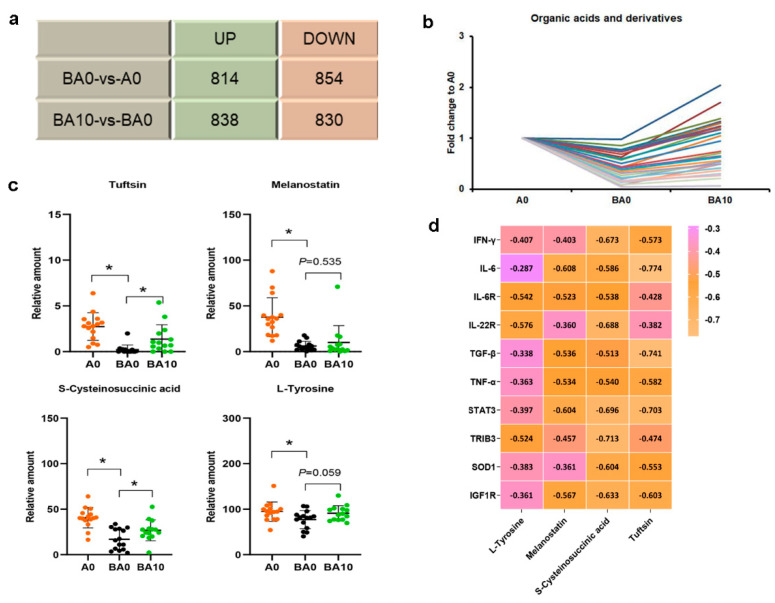
AOS 10 mg/kg benefited blood metabolism. (**a**). Summary of the changed metabolites in the different compared groups. (**b**). Fold-change in A0 group of organic acids and derivatives. The *y*-axis represents the fold-change in A0. The *x*-axis represents the treatment. (**c**). The metabolites changed in each treatment. The *y*-axis represents the relative abundance of metabolites. The *x*-axis represents the treatment. n = 15, * *p* < 0.05 (**d**). Correlation between metabolites and liver injury proteins.

**Table 1 ijms-24-03097-t001:** Primers used for q-RT-PCR.

Genes	Sequences (5’–3’)	Fragment Size (bp)	Accession No.
*β-Actin*	F: TCGTGGGCCGCTCTAGGCAC	255	NM_007393.3
	R: TGGCCTTAGGGTTCAGGGGGG		
*GAPDH*	F: AACAGCAACTCCCACTCTTC	111	NM_001289726.1
	R: CCTGTTGCTGTAGCCGTATT		
*IFNAR1*	F: CACGGTCGCTGTAGAAGTAAAG	104	NM_010508.2
	R: TCTCCTCCTCTTCGTTGGAATA		
*IFNAR2*	F: GACTTAAGAGCTGAGCAGGATG	106	NM_001110498.1
	R: AGACGGTGTGATAGTCTCTAGG		
*Tgfb3*	F: CGCTACATAGGTGGCAAGAA	108	NM_009368.3
	R: CAAGTTGGACTCTCTCCTCAAC		
*CD86*	F: CCTGGAAAGGTCTGGAGAATG	110	NM_019388.3
	R: GGCAGATCAGTCCTTCCATAAA		
*Tlr4*	F: GAGCAAACAGCAGAGGAAGA	111	NM_021297.3
	R: CCAGGTGAGCTGTAGCATTTA		
*Mrc1*	F: GGCGAGCATCAAGAGTAAAGA	98	NM_008625.2
	R: CATAGGTCAGTCCCAACCAAA		
*Ccl2*	F: GCTCAGCCAGATGCAGTTA	104	NM_011333.3
	R: CTGCTGGTGATCCTCTTGTAG		
*Egr1*	F: CGTCCTGTTCCCTTTGACTT	122	NM_007913.5
	R: GCATGTGATGGAGAGGATACTG		
*JUN*	F: CCAGACTGTACACCAGAAGATG	102	NM_010591.2
	R: CAACCAAAGTGTCTGCTTTCC		
*Ifit1*	F: CAGGATATTCACCTCCGCTATG	117	NM_008331.3
	R: CCTCCAAGCAAAGGACTTCT		
*Tlr7*	F: GTACCAAGAGGCTGCAGATTAG	135	NM_001290755.1
	R: CCTCAAGGCTCAGAAGATGTAAG		
*Ifi44*	F: GCTGGGAAGTCTAGCTTTGT	110	NM_001370771.1
	R: GTCCTGTACTTGTCAGAGATTCC		

**Table 2 ijms-24-03097-t002:** Antibodies used for WB and IHF.

Antibodies	Cat. No.	Company (accessed on 23 August 2020)
*β-Actin*	D110001-0200	www.sangon.com.cn
*IFN-γ*	bs-0481R	www.bioss.com.cn
*IL-6*	bs-0782R	www.bioss.com.cn
*IL-6R*	bs-23660R	www.bioss.com.cn
*IL-22R*	bs-2624R	www.bioss.com.cn
*TGF-β1*	bsm-33345M	www.bioss.com.cn
*TNF-α*	bs-10802R	www.bioss.com.cn
*STAT3*	bs-55208R	www.bioss.com.cn
*TRIB3*	bs-7538R	www.bioss.com.cn
*GPX1*	bs-3882R	www.bioss.com.cn
*SOD1*	bs-10216R	www.bioss.com.cn
*IGF1R*	bs-4985R	www.bioss.com.cn

## Data Availability

The liver RNA-seq raw data presented in this study were deposited in NCBI’s Gene Expression Omnibus under accession number GSE143245.

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
