# Peer review of "Alginate Oligosaccharides Repair Liver Injury by Improving Anti-Inflammatory Capacity in a Busulfan-Induced Mouse Model"

_ijms, 2023, doi:10.3390/ijms24043097_

Round 1

Reviewer 1 Report

First of all, I would like to thank you for the opportunity to review this study. 

I believe that the subject matter of this research is suitable for this journal, however I think that a couple of things need to be modified. 

Regarding the contextualisation of the study, I think it is necessary to include a greater number of current bibliographical references, both in the introduction and in the discussion. Also, add what this research contributes to the field of study. 

Likewise, the presentation of the article should be improved as a whole. Not even the font has been respected and everything seems to be placed in a chaotic way.

Reviewer 2 Report

1- You didn't mention the source of Alginate oligosaccharides.

2- AOS dose (10mg/kg) preparation is unclear. 

3- On what basis did you choose the AOS dose ?

4- Many abbreviations need to be clarified.

5- the supplementary material should be more organized, many data could be omitted and summarize the most crucial ones in a tabular form.

Round 2

Reviewer 1 Report

Thank you very much for reviewing the research. In this case I consider that everything I quoted previously has been considerably improved. I therefore give the go-ahead for the publication of this research.